# Fermentative Liberation of Ellagic Acid from Walnut Press Cake Ellagitannins

**DOI:** 10.3390/foods11193102

**Published:** 2022-10-05

**Authors:** Wolfram M. Brück, Víctor Daniel Díaz Escobar, Lindsay Droz-dit-Busset, Martine Baudin, Nancy Nicolet, Wilfried Andlauer

**Affiliations:** 1Institute of Life Technologies, University of Applied Sciences and Arts Western Switzerland Valais, Rue de l’Industrie 19, 1950 Sion, Switzerland; 2School of Agriculture, Forestry & Food Science, Bern University of Applied Sciences, 3052 Zollikofen, Switzerland

**Keywords:** nut press cake, ellagitannins, ellagic acid, fermentation

## Abstract

Oil is extracted from walnut leaves behind large quantities of defatted press cake that is still rich in valuable nutrients. *Aspergillus oryzae* and *Rhizopus oligosporus*, two molds traditionally used in Asia, have the necessary enzymes to use the nutrients in the walnut press cake. Walnuts and the press cake contain ellagitannins, known as precursors for ellagic acid and urolithins. In this study, experiments to optimize the solid-state fermentation of walnut press cake were performed in order to liberate ellagic acid from ellagitannins. Extracts of fermented products were then analyzed with an HPLC-DAD to measure the liberation of ellagic acid from ellagitannins. Good growth of *R. oligosporus* and *A. oryzae* mycelia on the walnut press cake was observed. A single mold culture was subjected to a hydration of 0.8 mL/g, an addition of 37.5 mmol/kg acetic acid (AA) and 1% NaCl, and an incubation temperature of 25 °C; these were observed to be good conditions for solid-state fermentation for walnut press cake. The highest ellagic acid concentration was obtained at 48 h. At 72 h, degradation dominated the liberation of ellagic acid.

## 1. Introduction

For thousands of years, fermentation has been used to preserve food, to improve taste, and to enhance digestion. Yet, fermentation provides food qualities that are far beyond the aforementioned properties. Recently, fermentation has received much attention as a means to increase the value of food by-products from the agri-food industry. Fermentative processes can create new economic value from agricultural and food industrial waste. Annually, one-third of the total food production is lost, which equals about 1.3 billion tons, of which 20% is from oilseeds [1]. About half of the total waste comes from cultivation and post-harvest losses, while the other half comes from processing, distribution, and final consumption [2]. For 2016, the European Commission estimated that the lost value from the waste of BP coming from the agri-food industry was 143,000 million euros [3]. 

The by-products from the agri-food industry can be classified as edible and non-edible. A non-edible by-product contains toxins, and thus cannot be used for human nutrition. Contrarily, edible by-products can be used for animal and human consumption due to their remaining nutritional value and the absence of toxins [2]. Furthermore, edible by-products can be processed further to serve as a source of proteins and antioxidants; they may also be used as substrates for producing enzymes, antibiotics, amino acids, vitamins, flavors, pigments, or surfactants [1].

In the last decade, there has been much interest from the food and cosmetic industry in the oil that comes from walnuts (*Junglas regia* L.), which is rich in bioactive compounds, such as polyunsaturated fatty acids or fat-soluble vitamins; moreover, the oil is easily extracted via cold pressing [4]. In 2019/2020, the oil production coming from seeds reached 580 million metric tons worldwide [1]. Concomitantly, large amounts of by-products in the form of press cake (pomace) were produced [1]. For instance, the oil extraction from walnuts uses twice the kernels in mass as the volume of oil produced, leaving behind vast quantities of walnut press cake (WPC) [5]. 

The protein content in nuts varies considerably. Together with peanuts, almonds, pistachios, and cashews, walnuts have the highest protein content, at around 20% or higher [6]. Therefore, WPC is an extremely rich source of protein. However, the resulting composition of WPC depends on the cultivar, provenance, agricultural practices, and the oil recovery process applied. Nevertheless, on average, WPC still contains oil (20–36%), depending on pressure conditions, substantial amounts of protein (30–42%) that become enriched during oil pressing, dietary fiber, phenolic compounds (ellagitannins), and minerals [5,7,8,9]. This interesting composition of WPC explains the high interest in using WPC as a nutritive and functional component in food products, or as a substrate for fermentation [1,5]. 

Fermentation has historically been used as a way of processing food for human consumption. Initially used for preservation, fermentation enhances the quality and functionality of food [10]. However, in modern times, techniques have improved capabilities to increase the productivity and efficiency of fermentation processes. Currently, there are two principal means of processing used: (I) submerged fermentation and (II) solid-state fermentation (SSF) [9]. SSF, also used in the present study, consists of using a solid substrate in the absence of liquid water where microorganisms can grow; in this way, the nutrients are maximally concentrated in the solid substrate [11,12]. Thus, the advantages of using SSF are high yields at low costs, high activity of microorganisms, low water consumption, little waste production, and higher resistance to contamination [13]. Nonetheless, this process has some shortcomings, making the use of SSF for industrial purposes challenging; some issues include scalability issues, heat accumulation, and difficulties with the regulation of growth parameters [11,14].

The most commonly used microorganisms in SSF are filamentous fungi. The parameter that most affects the growth of microorganisms in SSF is the fermentation temperature. However, growth is also affected by the moisture content, the chemical composition and particle size of the substrate, the height of the substrate layer, oxygen accessibility, the initial concentration of spores, the distribution of microorganisms in the substrate, and the age of the microorganisms [13,15].

For this study, fungi of the genera *Aspergillus* and *Rhizopus* were used. It is known that these two genera of fungi can use a broad range of substrates to produce enzymes which are active at lower pH and high temperatures, resulting in the liberation of phenolic compounds of biological interest, such as ellagic acid (EA) [13]. *Rhizopus oligosporus* exhibits strong lipolytic and proteolytic activities, while *Aspergillus oryzae* is unique in that it can ferment both carbohydrate-rich and protein-rich foods.

Phenolic compounds such as EA have gained considerable attention, as they are the most abundant antioxidants in the human diet [16]. EA, a hetero-tetracyclic compound that is produced from the dimerization of gallic acid, is a polyphenol that is found in countless fruits, vegetables, and nuts [17]. It is found in the cell vacuole and cell wall in free form, or bound to sugar molecules [18]. EA can be industrially liberated from ellagitannins via acidic or alkaline hydrolysis. However, the liberation of EA comes with many drawbacks, such as high production costs, generation of secondary products, and low yield [19]. Therefore, the liberation of EA using a fermentative process represents a valuable alternative that is in line with green biotechnology and the United Nations Sustainable Development Goals [11,13,16,20]. 

EA, besides being an antioxidant, provides several health benefits; anti-mutagenic, antitumor, anti-inflammatory, and antimicrobial activities have been reported [1,4,17]. These beneficial effects are mainly attributed to EA metabolites called urolithins, which are formed by colonic microbiota [21,22].

Thus, the aim of the present study was to apply SSF on WPC, using the fungi *A. oryzae* and *R. oligosporus*, with a particular interest in the transformation of ellagitannins into EA. From this study, a fermented food-quality WPC that was characterized for its EA content was expected. 

## 2. Materials and Methods

### 2.1. Walnut Press Cake (WPC)

Four and a half kg of WPC was provided by Huilerie Pré Girard, Pompaples (Switzerland), as a dry powder. The WPC was homogenized and separated into portions of 500 g and stored under vacuum at 4 °C until used for fermentation experiments. 

Characterization of WPC before fermentation showed a dry matter content of 95.45% (Halogen Balance, Mettler-Toledo GmbH, Columbus, OH, USA). Based on DM, the WPC showed a protein content of 29.7% (Kjeldahl), a fat content of 37.6% (Soxhlet), a mineral content (ash) of 4.1%, and carbohydrate and dietary fiber content of 28.6% (calculated). The most frequent particle size was 60 μm, with 50% of the particles being smaller than 340 μm, and 90 % of particles being smaller than 1050 μm (Camsizer^®^ XT, Retsch Technologie, Haan, Germany).

### 2.2. Hydration Solution

A hydration solution (HS) was prepared that contained 2.5 g/L of NaNO_3_ (Fluka, Aesch, Switzerland); 1 g/L of KH_2_PO_4_ (Sigma-Aldrich, Buchs, Switzerland); 0.5 g/L of KCL (Sigma-Aldrich, Buchs, Switzerland); and 0.5 g/L of MgSO_4_ (Fluka, Aesch, Switzerland). 

### 2.3. Acidification Solution

An aqueous acetic acid (AA, Nutrex, Lyss, Switzerland) solution with a concentration of 45 g/L (pH of 2.5) was used to acidify the WPC before fermentation. 

### 2.4. Microorganisms

A tempeh starter culture containing freeze dried *Rhizopus oligosporus* spores in rice flour was obtained from MakrobiotikVersand (Nordhackstedt, Germany). *R. oligosporus* has previously been shown to have lipase, endoglucanase, endoxylanase, and aminopeptidase activities in soybean tempeh [23]. An active culture of *Aspergillus oryzae* DSM 1863 was obtained from DSMZ (Braunschweig, Germany). Aspergillus oryzae DSM 1863 has shown great potential for the secretion of hydrolytic enzymes and organic acids, and has an increased tolerance to liquid pyrolysis. [24]. The microorganisms were selected as a result of their commercial availability, robustness, and use in the food industry to produce tempeh. 

*A. oryzae* DSM 1863 was proliferated from conidia on potato dextrose agar (Biolife, Milan, Italy) for 7 d at 30 °C [25,26]. Before WPC inoculation, a suspension of spores of *A. oryzae* DSM 1863 (3.6 × 10^7^ cfu/mL) was prepared using an isotonic medium of maximum recovery diluent made of 5.0 g/L of peptic digest of animal tissue, and 42.5 g/L of NaCl (Biolife, Milan, Italy). The concentration of spores was measured using a Neubauer counting chamber [26].

### 2.5. Preparation of WPC 

The HS was used for the initial hydration of WPC by adding 80 mL of standard solution for each 100 g of dry WPC at room temperature, and mixing (Stomacher^®^, Worthing, United Kingdom) for 10 min in a sterile plastic bag. From the acidification solution, 5 mL were added per 100 g of dry WPC. NaCl was added to the hydrated and acidified WPC at a concentration of 1 g/100 g of WPC.

### 2.6. WPC Inoculation

The prepared WPC (hydrated, acidified, and salted) was inoculated with tempeh starter that contained *R. oligosporus*, following the manufacturer’s instructions. For 100 g of WPC, 0.76 g of tempeh starter was dissolved in 7.6 mL of distilled water. The dissolved tempeh starter (0.1 g/mL) was then added to the mixing bag containing the conditioned WPC (100 g) and mixed for 10 min at room temperature to yield a spore concentration of 3.6 × 10^5^ spores/g of WPC.

Another incubation was made with *A. oryzae* using a 3.6 × 10^7^ cfu/mL spore suspension, as described above. To 100 g of the conditioned WPC, 10 mL of a solution containing 3.6 × 10^6^ spores/mL were added and mixed for 10 min at room temperature to yield a spore concentration of 3.6 × 10^5^ spores/g of WPC. 

As a third experimental run, both microorganisms, *A. oryzae* and *R. oligosporus*, were used in a co-culture. Amounts of 3.6 × 10^5^ spores/g of WPC each of *R. oligosporus* and *A. oryzae* were added for these experiments. 

### 2.7. Fermentation

Disks with a diameter of 5 cm, a weight of 30 g, and a thickness of 1 cm, were made from hydrated acidified and inoculated WPC. The disks were placed in a glass mold and were separated spatially from each other so that they did not touch. The glass mold was covered with a perforated plastic film and placed in an incubator at 25 °C.

### 2.8. Sampling

Samples of the fermenting WPC were taken directly after inoculation (0 h), after 24 h, 48 h, and 72 h. For the sampling, approximately 3 g of the fermenting WPC disks were taken with a spatula at randomized locations and added to a 15-milliliter falcon tube. Tubes were stored at −20 °C until further analysis. 

### 2.9. Sample Preparation

Before extraction, about 2 g of frozen WPC samples were pre-dried for 24 h at 50 °C in an oven [7]. The water content and the dry matter content of the previously dried samples of fermented WPC were determined with a Halogen Balance HE73 Moisture analyzer (Mettler-Toledo GmbH, Greifensee, Switzerland).

### 2.10. EA Extraction

An volume of 5 mL of an acetone:water (80:20, *v*:*v*) solution were added to 200 mg of previously dehydrated samples of fermented WPC, and extracted in an ultrasonic bath for 15 min at room temperature. After that, samples were centrifugated at 3000 rpm for 10 min at room temperature [27]. A volume of 500 μL of supernatant was collected. The remaining pellet was extracted a second time using the same procedure. Again, 500 μL of supernatant was collected and combined with the first 500 μL. The combined extracts were filtered with NM Chromaphil^®^Xtra 0.45 μm and used for HPLC-DAD analysis [17,28]. 

### 2.11. HPLC-DAD Analysis

For quantification of EA, an HPLC-DAD method was applied (HPLC Agilent technologies, 1220 Infinity LC, Santa Clara, USA) with a C18 column Core-shell (Phenomexe, 50 × 2.1 mm, Kinetex 2.6 μm, EVO C18 100Å, Los Angeles, CA, USA). Eluant A consisted of ultrapure water that was acidified with 1% formic acid. Eluant B was acetonitrile with 1% formic acid. The column temperature was maintained at 35 °C. The injected sample volume was 5 μL. The gradient applied was 0–5 min, 5% B; 5–12 min, 5–60% B; 12–20 min, 60–95% B; 20–20.1 min, 95–5% B. The flow rate was maintained at 1 mL/min, and absorption was monitored at 260 nm [29].

### 2.12. Calibration Curve 

A commercial sample of EA (E2250-5G) obtained from Sigma-Aldrich (Buchs, Switzerland) was used as a standard for external calibration. A stock solution was prepared using 25 mg of EA dissolved in 25 mL of dimethyl sulfoxide (DMSO, Sigma-Aldrich, Buchs, Switzerland) to obtain a concentration of 1000 mg/mL. The EA stock solution was diluted in a mixture of eluants A and B (1 + 1), in order to obtain concentrations of 100, 75, 50, 25, and 10 mg/mL. 

### 2.13. Statistics

A simple ANOVA was used with a significance level of 0.05 to seek significant differences between EA concentrations obtained from the treatments, and at specific sample collection times. 

## 3. Results and Discussion

Solid-state fermentation for the liberation of ellagic acid in WPC using *R. oligosporus* and *A. oryzae*, either alone or in coculture, was performed. For experiments with *R. oligosporus*, a commercial tempeh starter culture was used. The production of tempeh is usually carried out using *R. oligosporus* on soybeans. *R. oligosporus* can degrade the cell wall of plants. In addition, it has proteolytic and lipolytic enzymes and can produce antibiotic substances against pathogenic Gram-negative bacteria. A second mold, *A. oryzae*, was also applied. It is used mainly to produce koji, sake, shochu, miso, and soy sauce. *A. oryzae* has multiple enzymatic activities with hydrolytic enzymes, such as proteases, to degrade proteins [30]. The SSF was performed using conditions that were identified as most appropriate to liberate EA.

For all fermentations, good growth of *R. oligosporus* and *A. oryzae* mycelia, alone or in coculture, was observed on the surface of the WPC tempeh disks after 48 h (Figure 1). The concentration of free EA in WPC is usually low, since EA is mainly bound in ellagitannins. In walnuts, EA is usually found at concentrations of about 600 μg/g [31], which is close to the values we obtained at time 0 h. Figure 2 shows a chromatogram of an water:acetone extract of walnut press cake at 260 nm. In order to identify the EA peak (Rt = 8.3 min), the samples were spiked with an EA reference solution. 

In all experiments, a steady increase in the concentration of EA was observed initially. However, a decrease in EA concentration was observed after longer fermentation times, presumably due to a degradation of EA by the fermenting fungi or by the endemic WPC microbiota (Table 1). Using sterilization with an autoclave was not a viable option because overheating the WPC causes a Maillard reaction accompanied by the appearance of off-flavors, compaction of the paste, and negligible liberation of EA. 

The presence of water during fermentation is a fundamental prerequisite; it influences growth, protein stability, production of metabolites, and has a role in substrate transport and as a reactant [32]. Generally, around 60 to 70% moisture content is optimal during SSF for fungi [15]. Hydration is a physical process that is related to permeability and colloidal properties [32]. Low hydration can cause growth disruption and an inability to interact with the substrate. On the other hand, high hydration can cause particle agglomeration, thus limiting oxygen availability [15]. 

Fungal metabolism and growth are related to the substrate’s water activity. Water activity is dependent on the water available; thus, a slight change in hydration may cause a significant effect on the metabolism of fungi [15]. Decreased water activity slows growth, while high water activity improves growth. However, accelerating effects on the metabolism may also cause stress, leading to degradative reactions [32]. It can be argued that too high an increase in the metabolism may be counterproductive, as EA could be degraded; hence optimal hydration is necessary for SSF, in order to prevent under- or overgrowth.

Temperature is an important parameter to control in SSF. In Indonesia, traditional tempeh fermentation is carried out at ambient temperature (~30 °C) [33]. Molds can survive a broad range of temperatures, from 20 to 55 °C. An optimum temperature will provide the best growth velocity and metabolite output, as mold enzymatic metabolism becomes most efficient [15]. The fastest mold development and the highest EA concentration in fermentations with *R. oligosporus* was observed at an incubation temperature of 25 °C after 48 h. This result could be related to the metabolic heat and increased incubation temperature, resulting in optimal conditions for growth and metabolic output for *R. oligosporus* during SSF of the WPC. In addition, heat can cause EA liberation reactions simultaneously as metabolic reactions, increasing EA concentration [15,33]. 

Degradation was observed after 72 h for most fermentations. It is known that the degradation of polyphenolic compounds such as EA can occur at high temperatures [28]. In our case, an increase in the incubation temperature on the fermenting WPC by the generated metabolic heat may have caused degradation of EA, either by metabolic stress and degradative enzymatic reactions from R*. oligosporus* and *A. oryzae*, or as a result of simple heat degradation.

Acidity and pH value are critical factors for the mycelial growth of *R. oligosporus* and *A. oryzae*, and the inhibition of contaminating bacteria [34]. The fungi can thrive in a broad range of pH, giving them the advantage of low bacterial competition [15]. In traditional tempeh fermentation using *R. oligosporus*, the initial pH is between 4.5–5.5, and increases to 6.5–7.0 as fermentation continues in mature tempeh [32]. However, in tempeh production, a low initial pH is obtained when lactic and acetic bacteria grow during the soaking of the soybeans [34]. Therefore, it was decided to add food-grade AA to the WPC before inoculation. 

A study that was conducted with rapeseeds found a window for optimal *R. oligosporus* growth and inhibition of bacteria that was achieved with 40–60 mmol AA per kg of substrate. The fastest growth and complete surface coverage after 24 h were achieved with a concentration of AA of 37.7 mmol/kg of substrate [34]. In our case, the AA solution added at a ratio of ml per 100 g of WPC led to 37.5 mmol of AA per kg of WPC.

The use of an organic acid had a notable effect on the speed and intensity of *R. oligosporus* mycelial growth. Acidity, and thus pH, directly impact the growth of the mold, increasing the accessibility and liberation of EA by increasing mold metabolism. However, the 37.5 mmol/kg AA solution appeared to have a slight edge on the liberation of EA and in taste. Thus, AA was selected as the optimal organic acid for WPC fermentation.

The salt concentration is known to suppress the growth of spoilage microorganisms and pathogens, while favoring the growth of halotolerant microorganisms that play an essential role in forming flavor molecules [35]. Salt was administered with a focus on taste and surface mycelia formation. The best-perceived taste was with 1% of salt and 37.5 mmol/kg of AA. However, the addition of salt caused a delay in the growth of *R. oligosporus* and *A. oryzae* mycelia, affecting the EA concentration at 48 h. It is known that high salt concentrations can inhibit the enzymatic activity of the mold, causing metabolic stress, consequently affecting mycelial growth, and either indirectly slowing the liberation of EA or promoting degradation [35]. 

The effects on EA liberation due to fermentation in WPC with *A. oryzae* and *R. oligosporus*, alone or in coculture, were studied in triplicate. There were no significant differences found in the EA concentrations when fermenting WPC with *A. oryzae* or *R. oligosporus* alone. However, the coculture showed significantly lower EA liberation at 48 h (*p* = 0.01) and 72 h compared to fermentation with *A. oryzae* or *R. oligosporus* alone.

The increase in EA concentration observed was comparable for all three fermentations after 24 h. After 24 h, differences began to appear in the liberation of EA. *A. oryzae* and *R. oligosporus* can liberate EA with similar efficacy until 48 h, when the concentration of EA seemed to either stabilize or began to degrade at 72 h. When the molds were in coculture, the EA concentrations were equal at 24 h to the single cultures, but changed thereafter. The coculture showed slow mycelial growth, indicating that there was competition between the molds for the substrate. A plateau in the accessibility to EA was reached with lower EA concentrations.

## 4. Conclusions

A fermentative process is a viable option to take advantage of food by-products from the agricultural industry. The SSF of WPC is an exemplary process for creating products with added nutritional value. For industrial upscaling, parameters such as water content, acidity, and temperature must be optimized. The liberated EA has been shown to be of nutritional value, and may serve as a precursor for beneficial urolithins. It has been shown that the digestion and microbial transformation of EA-containing food releases urolithins [21]. Unfortunately, not all humans are able to produce the most health-promising form, urolithin A. Therefore, another possible approach would be to apply microbial transformation of the fermented WPC to produce urolithin A directly in the food, rendering the product valuable for non-producers of urolithin A.

## Figures and Tables

**Figure 1 foods-11-03102-f001:**
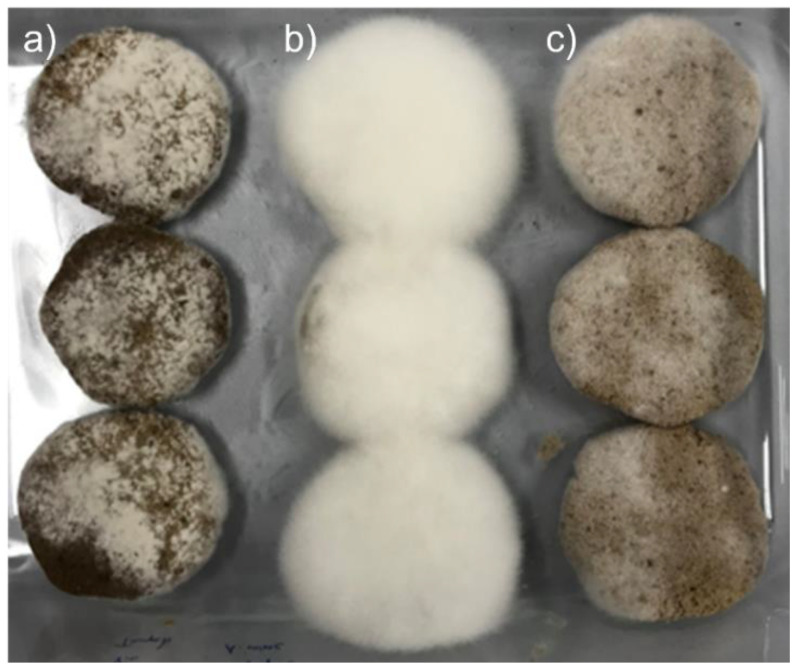
Growth of mold mycelia on WPC after 48 h: (**a**) *A. oryzae*, (**b**) *R. oligosporus*, and (**c**) coculture of *A. oryzae* and *R. oligosporus*.

**Figure 2 foods-11-03102-f002:**
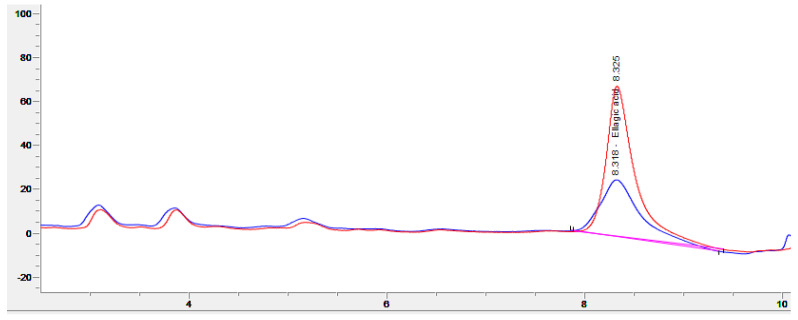
Chromatogram (260 nm) of an extract from walnut press cake (blue), and the same sample spiked with ellagic acid (red).

**Table 1 foods-11-03102-t001:** EA liberation from SSF of WPC using *A. oryzae*, *R. oligosporus*, and in coculture, n = 3 (±SD).

Treatment	0 h	24 h	48 h	72 h
*Aspergillus oryzae*		1.13 ± 0.18 mg/g	2.57 ± 0.44 mg/g	2.31 ± 0.32 mg/g
*Rhizopus oligosporus*	0.42 ± 0.01 mg/g	1.19 ± 0.43 mg/g	2.71 ± 0.18 mg/g	2.53 ± 0.69 mg/g
*A. oryzae* and *R. oligosporus*		1.24 ± 0.33 mg/g	1.78 ± 0.03 mg/g	1.78 ± 0.23 mg/g

## Data Availability

The data presented in this study are available on request from the corresponding author.

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
