# Peer review of "Fermentative Liberation of Ellagic Acid from Walnut Press Cake Ellagitannins"

_foods, 2022, doi:10.3390/foods11193102_

Round 1

Reviewer 1 Report

Previous studies have shown that the acid environment of gastric juice and the glucosidase produced by the epithelial cells and intestinal flora in the intestinal mucosa can hydrolyze ellagitannins to produce ellagic acid. Please further explain the necessity and innovation of this step transformation in vitro.

Line 217-219, This article describes that "autoclaving can cause Maillard reaction, so it is not recommended to use this method for sterilization", so please explain here how sterilization is carried out in this article?

Line 220, Preferably, the consumption of substrate ellagitannin should be shown in this table.

Line 320-322, Please check whether the citation format of reference [3] is correct, and [3] and [4] all cite the document "Estimates of European food waste levels." Please make a revision.

Author Response

We appreciate the reviewer’s comments and believe to have answered the comments sufficiently below for the paper to be reconsidered for publication in Foods:

Previous studies have shown that the acid environment of gastric juice and the glucosidase produced by the epithelial cells and intestinal flora in the intestinal mucosa can hydrolyze ellagitannins to produce ellagic acid. Please further explain the necessity and innovation of this step transformation in vitro.

  • This step transformation was used as a means to valorize press cake from the nut oil industry by producing a Tempeh-like product from using Aspergillus and Our final aim is to form metabolites of ellagic acid, the urolithins, in the fermented food. Not all persons are able to form the health-providing urolithinin A. Therefore, the liberation of ellagic acid in food is pertinent to provide the precursor for urolithin formation.

Line 217-219, This article describes that "autoclaving can cause Maillard reaction, so it is not recommended to use this method for sterilization", so please explain here how sterilization is carried out in this article?

  • The product was not sterilized. However, due to the quantities of starter culture added, the intrinsic microorganisms in the raw material only contribute negligibly to the fermentation.

Line 220, Preferably, the consumption of substrate ellagitannin should be shown in this table.

  • We are aware of the advantage to follow the degradation of ellagitannins. However, there is more than one ellagic acid precursor and we are interested in ellagic acid liberation.

Line 320-322, Please check whether the citation format of reference [3] is correct, and [3] and [4] all cite the document "Estimates of European food waste levels." Please make a revision.

  • This has been corrected (missing reference added)

Reviewer 2 Report

Review Manuscript ID: foods-1878315, entitled “Fermentative Liberation of Ellagic Acid from Nut Press Cake Ellagitannins.”  

This is a very good paper indicating the experiments to optimize solid-state fermentation of walnut press cake, with the help of the molds Aspergillus oryzae and Rhizopus oligosporus, in a particular interest regarding the transformation of ellagitannins into EA (ellagic acid), in the context that large quantities of defatted press cake are still rich in valuable nutrients are disposed of as by-products. The purpose of the research is evident;

Some specific comments:

  • Line 2:  Regarding the Title of the paper, I suggest changing the “Nut” into “Walnut” because the walnut is the subject mentioned at point 2: “2. Materials and Methods

Walnut press cake (WPC) 

Four and a half kg WPC was provided by Huilerie Pré Girard, Pompaples (Switzer-104 land) as a dry powder.” In this context that the title must be in line with the subject of the research. After, I suggest rephrasing the title for proper understanding.

  • Line 56-58:WPC still contains oil (20-36%), 56 depending on pressure conditions, quite a lot of proteins (30-42%), enriched during oil 57 pressing, dietary fibers, phenolic compounds (ellagitannins), and minerals [5,7]”. I consider that only two references for this voluble statement is not convincing and recommend the following reference in addition: https://doi.org/10.3390/molecules25092214 (especially for quality oil (fatty acids) phenolic compounds and minerals and also for volatile profile composition of walnut oilcake powder)
  • Line 110: “carbohydrate and dietary fiber content of 28.6% (calculated)”. I recommend specifying the calculation formula, even if it is a common one
  • Line 197; Line 201; Line 206; and so on: “A. oryzae, R. oligosporus” must be Italic in all pages; authors are kindly invited to modify the entire document after line 197, where applicable.
  • Results and Discussion are logical and orderly blend with supporting of 1 table and 1 figure
  • References demonstrate the work of others in the field of the presented study, but some statements can be strengthened, as mentioned above. Regarding the reference, no 2 (Line 318), in the original article the Names of autors start with last name and second is First Name, (Petraru Ancuța and Amariei Sonia), so is required to check and if you consider to switch the name in References list (ex: Petraru A., Amariei S.)

Author Response

We appreciate the reviewer’s comments and believe to have answered the comments sufficiently below for the paper to be reconsidered for publication in Foods:

This is a very good paper indicating the experiments to optimize solid-state fermentation of walnut press cake, with the help of the molds Aspergillus oryzae and Rhizopus oligosporus, in a particular interest regarding the transformation of ellagitannins into EA (ellagic acid), in the context that large quantities of defatted press cake are still rich in valuable nutrients are disposed of as by-products. The purpose of the research is evident;

Some specific comments:

Line 2:  Regarding the Title of the paper, I suggest changing the “Nut” into “Walnut” because the walnut is the subject mentioned at point 2: “2. Materials and Methods

Walnut press cake (WPC)

Four and a half kg WPC was provided by Huilerie Pré Girard, Pompaples (Switzerland) as a dry powder.” In this context that the title must be in line with the subject of the research. After, I suggest rephrasing the title for proper understanding.

  • This has been changed.

Line 56-58: “WPC still contains oil (20-36%), 56 depending on pressure conditions, quite a lot of proteins (30-42%), enriched during oil 57 pressing, dietary fibers, phenolic compounds (ellagitannins), and minerals [5,7]”. I consider that only two references for this voluble statement is not convincing and recommend the following reference in addition: https://doi.org/10.3390/molecules25092214 (especially for quality oil (fatty acids) phenolic compounds and minerals and also for volatile profile composition of walnut oilcake powder)

  • The paper has been added

Line 110: “carbohydrate and dietary fiber content of 28.6% (calculated)”. I recommend specifying the calculation formula, even if it is a common one

  • The walnut press cake has been analyzed for its content in protein, fat, minerals and water. The difference to 100% was attributed to fiber and carbohydrates.

Line 197; Line 201; Line 206; and so on: “A. oryzae, R. oligosporus” must be Italic in all pages; authors are kindly invited to modify the entire document after line 197, where applicable.

  • This has been corrected throughout the manuscript.

Results and Discussion are logical and orderly blend with supporting of 1 table and 1 figure

References demonstrate the work of others in the field of the presented study, but some statements can be strengthened, as mentioned above. Regarding the reference, no 2 (Line 318), in the original article the Names of autors start with last name and second is First Name, (Petraru Ancuța and Amariei Sonia), so is required to check and if you consider to switch the name in References list (ex: Petraru A., Amariei S.)

  • This has been corrected in the manuscript references.

Reviewer 3 Report

The manuscript is of interest of the journal and does have novelty. In spite of that, it has a few issues that need to be addressed. There is a poor use of tables and figures, since there´s just one table and a single photo, maybe at least an HPLC chromatogram, or some other figure could have been included.

Most of the articles studying polyphenol liberation include some general characterisation of the samples, such as total phenolics or total antioxidants of the extracts prepared by fermentation. I consider those determinations should be necessary.

L79: “For this study, fungi of the genera Aspergillus and Rhizopus are used. It is known that 79 these two genera of fungus can use a broad range of substrates and producing enzymes 80 which are active at lower pH and high temperatures, resulting in the liberation of phenolic 81 compounds of biological interest, like ellagic acid (EA) [11].”

Have authors determine the main enzyme/s produced, that could be related to phenolic liberation? It could help provide insights in the conditions that were required for the process to work. Could authors at least name the main enzymes produced by these microorganisms related to phenolic liberation according to literature provided?

Have authors have some hypothesis on why EA production was according to them

L85: “Phenolic compounds such as EA have gained considerable attention as they are the 85 most abundant antioxidants in the human diet [14]. EA, a hetero-tetra-cyclic compound 86 coming from the dimerization of gallic acid is a polyphenol found in countless fruits, veg-87 etables, and nuts [15]. It is found in the cell vacuole and cell wall in free form or bound to 88 sugar molecules [16]. EA can be industrially liberated from ellagitannins by acid or alka-89 line hydrolysis. However, the liberation of EA comes with many drawbacks such as high 90 production cost, generation of secondary products, and low yield [17]. Therefore, the lib-91 eration of EA by a fermentative process represents a valuable alternative in line with green 92 biotechnology and United Nations Sustainable Development Goals [9,11,15, 31].”

Since authors are producing EA from tannins, by fermentation. Have authors evaluated if the degradation of tannins stops at EA? Or does go on until gallic acid in other conditions? The degradation reported after 48h of fermentation, is it due to the production of gallic acid from EA? Is the HPLC method able to discriminate EA from gallic acid? Have authors detected gallic acid in the fermentation treatments?

L 94 “EA, besides being an antioxidant, provides several health benefits. Anti-mutagenic, 94 antitumor, anti-inflammatory, and antimicrobial activities have been reported [1,4,15]. 95 These beneficial effects are mainly attributed to EA metabolites called urolithins, formed 96 by colonic microbiota [18,32].”

Gallic acid has interesting properties and was probably liberated in part as well.

L111 was WPC ground to this particular size or was it used as obtained and why? A comment could be useful.

L157: Samples of the fermenting WPC were taken directly after inoculation (0 h). How was this blank prepared? One sample taken from each of the three treatments, thus, a triplicate? It is not clear, as in the table just one result shows. If sample is taken AFTER inoculation, the table should show 3 results, instead of one?

L174: in the HPLC-DAD method for quantification of EA, an HPLC-DAD method was applied where the absorption was monitored at 260 nm. Given that the UV maxima of EA is about 277 nm, and gallic acid is about that wavelength as well, even if the value is close, the next time the authors can use a higher value to diminish the matrix impurities overlap on the quantification. That is, if the intention is to quantify EA solely.

L209: “in walnuts, EA is usually found at concentrations of about 600 µg/g [25], which is close to our values at time 0 h”

It was not completely clear to me how blanks were prepared and how are them the same value as the time 0 h.

L280: “There were no significant differ-280ences in the EA concentrations when fermenting the WPC with A. oryzae or R. oligospo-281rus alone. However, the coculture showed lower EA liberation at 48 h significantly282(p=0.01) and 72 h, compared to fermentation with A. oryzae or R. oligosporus alone.”

Have authors established hypothesis, reasons why? It is a trend nowadays to use microorganism consortia for best results, in that sense, are these two not compatible?

L285: “differences started to appear” would be better.

Given the few references that can be found on walnut cake, it is surprising that the authors did not cite the work of:

Bakkalbaşı, E. Oxidative Stability of Enriched Walnut Oil with Phenolic Extracts from Walnut Press-Cake under Accelerated Oxidation Conditions and the Effect of Ultrasound Treatment. Food Measure 2019, 13, 43–50, doi:10.1007/s11694-018-9917-y.

The article is very much related as it explores “the effect of phenolic extract obtained from walnut press-cake on oxidative stability of walnut oil was evaluated. Besides this, the oxidative stability of ultrasound applied walnut oil with added press-cake, as a valuable source of phenolic compounds”

Author Response

We appreciate the reviewer’s comments and believe to have answered the comments sufficiently below for the paper to be reconsidered for publication in Foods:

The manuscript is of interest of the journal and does have novelty. In spite of that, it has a few issues that need to be addressed. There is a poor use of tables and figures, since there´s just one table and a single photo, maybe at least an HPLC chromatogram, or some other figure could have been included.

  • HPLC chromatogram of a walnut press cake extract has been added (Figure 2).

Most of the articles studying polyphenol liberation include some general characterisation of the samples, such as total phenolics or total antioxidants of the extracts prepared by fermentation. I consider those determinations should be necessary.

  • In this study, only ellagic acid as possible precursor for urolithins is of interest. Ellagic acid is one of numerous polyphenols or antioxidants in walnut press cake. There, values of total phenolics or total antioxidants are not productive.

L79: “For this study, fungi of the genera Aspergillus and Rhizopus are used. It is known that these two genera of fungus can use a broad range of substrates and producing enzymes which are active at lower pH and high temperatures, resulting in the liberation of phenolic compounds of biological interest, like ellagic acid (EA) [11].”

Have authors determine the main enzyme/s produced, that could be related to phenolic liberation? It could help provide insights in the conditions that were required for the process to work. Could authors at least name the main enzymes produced by these microorganisms related to phenolic liberation according to literature provided?

  • We thank the reviewer’s valuable comment. We have stated the exact strain of oryzae (Aspergillus oryzae DSM 1863) in the manuscript which has previously been isolated from Indonesian food fermentations. The Rhizopus oligosporus used was a commercial started culture for Tempeh. Both strains were selected for their commercial availability, robustness, and use in the food industry to produce Tempeh. Aspergillus oryzae DSM 1863 has shown great potential for the secretion of hydrolytic enzymes and organic acids and have an increased tolerance to liquid pyrolysis. (Kubisch C, Kövilein A, Aliyu H, Ochsenreither K. RNA-Seq Based Transcriptome Analysis of Aspergillus oryzae DSM 1863 Grown on Glucose, Acetate and an Aqueous Condensate from the Fast Pyrolysis of Wheat Straw. J Fungi (Basel). 2022 Jul 23;8(8):765. doi: 10.3390/jof8080765.). R. oligosporus has previously shown to have  lipase, endoglucanase, endoxylanase, and aminopeptidase activities (Varzakas, T. Rhizopus oligosporus mycelial penetration and enzyme diffusion in soya bean tempe. Process Biochemistry 1998, 33(7): 741-747. https://doi.org/10.1016/S0032-9592(98)00044-2). Both papers have now been included in the manuscript.

Have authors have some hypothesis on why EA production was according to them

L85: “Phenolic compounds such as EA have gained considerable attention as they are the most abundant antioxidants in the human diet [14]. EA, a hetero-tetra-cyclic compound coming from the dimerization of gallic acid is a polyphenol found in countless fruits, vegetables, and nuts [15]. It is found in the cell vacuole and cell wall in free form or bound to sugar molecules [16]. EA can be industrially liberated from ellagitannins by acid or alkaline hydrolysis. However, the liberation of EA comes with many drawbacks such as high production cost, generation of secondary products, and low yield [17]. Therefore, the liberation of EA by a fermentative process represents a valuable alternative in line with green biotechnology and United Nations Sustainable Development Goals [9,11,15, 31].”

Since authors are producing EA from tannins, by fermentation. Have authors evaluated if the degradation of tannins stops at EA? Or does go on until gallic acid in other conditions? The degradation reported after 48h of fermentation, is it due to the production of gallic acid from EA? Is the HPLC method able to discriminate EA from gallic acid? Have authors detected gallic acid in the fermentation treatments?

  • We thank the reviewer’s valuable comment. The authors evaluated the structure of ellagic acid carefully and came to the conclusion that, even though gallic acid is contained within the structure of ellagic acid, it would be more likely for microorganisms to open the structure at easier accessible point such as the ester bondings of ellagic acid. Gallic acid was not quantified in the fermentation treatments, since it is not of interest as precursor for urolithins.

L 94 “EA, besides being an antioxidant, provides several health benefits. Anti-mutagenic, antitumor, anti-inflammatory, and antimicrobial activities have been reported [1,4,15]. 95 These beneficial effects are mainly attributed to EA metabolites called urolithins, formed by colonic microbiota [18,32].”

Gallic acid has interesting properties and was probably liberated in part as well.

  • Please see comment above.

L111 was WPC ground to this particular size or was it used as obtained and why? A comment could be useful.

  • The WPC was obtained from a commercial mill and used as is. The WPC was already in a fine powder and upon evaluation it was deemed unnecessary to further sieve the product into a particular size.

L157: Samples of the fermenting WPC were taken directly after inoculation (0 h). How was this blank prepared? One sample taken from each of the three treatments, thus, a triplicate? It is not clear, as in the table just one result shows. If sample is taken AFTER inoculation, the table should show 3 results, instead of one?

  • The walnut press cake was prepared like described in the M&M section “Preparation of WPC”. The analysis to get time 0 h was done directly after the inoculation in triplicate. All incubations underwent the same pretreatment, including the added volume of spore solutions. Therefore, all incubations are at the same concentration at time 0 h. Hence only one result is shown.

L174: in the HPLC-DAD method for quantification of EA, an HPLC-DAD method was applied where the absorption was monitored at 260 nm. Given that the UV maxima of EA is about 277 nm, and gallic acid is about that wavelength as well, even if the value is close, the next time the authors can use a higher value to diminish the matrix impurities overlap on the quantification. That is, if the intention is to quantify EA solely.

  • For this study, we used a method, which is applied also for the analysis of other phenolic compounds, without any modification. Therefore, we quantified at 260 nm. We thank the reviewer for this valuable advice.

L209: “in walnuts, EA is usually found at concentrations of about 600 µg/g [25], which is close to our values at time 0 h”

It was not completely clear to me how blanks were prepared and how are them the same value as the time 0 h.

  • The walnut press cake was prepared like described in the M&M section “Preparation of WPC”. The analysis to get time 0 h was done directly after the inoculation. The values stated in Line 209 are from a reference [25]. We have detected 420 µg/g in WPC at T0 as stated in Table 1.

L280: “There were no significant differences in the EA concentrations when fermenting the WPC with A. oryzae or R. oligosporus alone. However, the coculture showed lower EA liberation at 48 h significantly282(p=0.01) and 72 h, compared to fermentation with A. oryzae or R. oligosporus alone.”

Have authors established hypothesis, reasons why? It is a trend nowadays to use microorganism consortia for best results, in that sense, are these two not compatible?

  • Coculturing has been found to trigger the biosynthesis of diverse secondary metabolites and enzymes of microorganisms including fungi. Furthermore, it has also shown to be useful for uncovering the mechanisms of fungal interspecific interactions and novel gene functions. Unfortunately, in our assays, it may be possible that the two molds were perhaps competitively inhibiting each other from efficiently using the available substrates to produce ellagic acid. However, most likely, these findings are multifactorial. In that sense, as the reviewer has kindly stated, the two organisms may not have been compatible and thus unable to work on co-culture. A critical review of co-cultures including their drawbacks has been published by Kapoore et al (2020): https://doi.org/10.1080/07388551.2021.1921691.

L285: “differences started to appear” would be better.

  • This has been changed in the text.

Given the few references that can be found on walnut cake, it is surprising that the authors did not cite the work of:

Bakkalbaşı, E. Oxidative Stability of Enriched Walnut Oil with Phenolic Extracts from Walnut Press-Cake under Accelerated Oxidation Conditions and the Effect of Ultrasound Treatment. Food Measure 2019, 13, 43–50, doi:10.1007/s11694-018-9917-y.

The article is very much related as it explores “the effect of phenolic extract obtained from walnut press-cake on oxidative stability of walnut oil was evaluated. Besides this, the oxidative stability of ultrasound applied walnut oil with added press-cake, as a valuable source of phenolic compounds”

  • The reference has been added where necessary.

Round 2

Reviewer 1 Report

I agree to accept this paper.

Reviewer 3 Report

The authors carried out an interesting work, concise to a modest goal.